# How does visual experience shape representations and transformations along the ventral stream?

*CCN Generative Adversarial Collaboration*

*Bedny, M., Kanwisher, N., Collignon, O., Yildirim, I., Saccone, E. J., Murty, A. P. R., Mattioni, S.*

**Scientific question:** A key puzzle in cognitive neuroscience concerns the contribution of innate predispositions versus lifetime experience to cortical function. Addressing this puzzle has implications that reach far, from plasticity of the neural hardware [1] to representations in the mind and their developmental trajectory [2], and even to building artificially intelligent systems [3,4]. Yet, this is a notoriously difficult topic to study empirically. We propose to tackle this issue in the context of the high-level 'visual' representations through neural, behavioral, and computational studies of individuals who are sighted and congenitally blind. Congenital blindness represents a uniquely tractable and rich model to study how innate substrate and atypical experience interact to shape the functional tuning of the brain. This work aspires to reveal the origins, including the representational and computational basis, of high-level visual representations by addressing the following questions: *How does visual experience impact representations and transformations along the ventral stream? How broad is the human brain's capacity to 'retool' in the face of 'atypical' experience?*

**Background:** Several decades of research in sighted individuals has documented object category-specific responses in the ventral occipito-temporal cortices (vOTC). Specific areas respond preferentially to faces, scenes and letters more strongly than other categories of objects (Fusiform Face Area (FFA) [5]; Parahippocampal Place Area (PPA) [6]; Visual Word Form Area (VWFA) [7]). These cortical representations are thought to support visual object recognition and identification for these categories (e.g., [8-10]). For example, the FFA is thought to represent holistically the shape of the face, facial features and the spacing between them [11-12]. The PPA represents the global, spatial layout of scenes [13,9]. The VWFA represents letters, letter combinations and words to support word recognition [7,8].

The relative contributions of innate predispositions and lifetime visual experience to the development of these regions is not well known. One view is that the location and representational content of these regions is determined by experience with low-level perceptual properties of different visual object categories (e.g., [14]). For example, the FFA arises in its typical location because of a general bias of that region to receive round and curvy visual input from the fovea [15,16]. Contrary to this idea, a number of studies of people who are born blind find analogous selectivity in the ventral stream to that of the sighted. In other words, neural responses to stimuli from domains of faces, places and written words are observed in blind people in similar locations within the ventral 'visual' stream to those of sighted people. For example, hearing words referring to places or place sounds elicit activity in the location of the PPA [17,18]. Haptically explored faces and Braille reading have activated regions in FFA- and VWFA-like locations, respectively [19,20]. These studies suggest that the localization of category-specific areas within the ventral stream may be in part innately determined, and specifically does not require visual experience [17,18].

On the other hand, these observations do not speak to the specific function of these regions in people born blind. These gaps in our understanding make the theoretical significance of the existing evidence unclear. First, it is not known exactly how similar the functions of ventral stream regions are across blind and sighted people. To what degree do these regions represent the same information and play a similar behavioral role, i.e., object perception? It remains possible that while these ventral stream areas show preferences for similar domains across sighted and blind people (i.e., FFA-people, PPA-places, VWFA-language), their representational content and behavioral role is quite different. From a behavioral perspective, blind and sighted people differ importantly in their object recognition behavior. For example, blind people do not use faces to recognize others, in part because it is socially awkward and inconvenient to touch strangers. Does the FFA-like area in people who are born blind still represent face shape and configuration of facial features in a holistic fashion, as in the sighted brain? Alternatively, does it support a similar computation but using different sensory input (e.g., auditory person recognition via voices)? Or does it represent something different still, like higher-level social or emotional information?

In line with the possibility that these neural representations are different in people who were born blind, another line of research suggests that the visual system of congenitally blind individuals, including the vOTC, takes on higher-cognitive functions that are computationally distant from the "original" role of vision [1]. For example, visual cortices of blind people are active during spoken sentence processing and sensitive to the grammatical complexity of sentences [21-23]. Other regions in the 'visual' cortex, including in the ventral stream, are active when blind participants solve spoken math equations [24]. Moreover, resting state connectivity between visual cortices and fronto-parietal networks increases in blindness [25]. Contrary to evidence of preserved function, these studies suggest that the blind visual cortex undergoes massive representational reorganization [1]. Whether the same or different portions of 'visual' cortex show high-order cognitive responses to math and language and also responses to tactile/audio faces, places and letters is not known.

**Challenge or controversy:** The crux of the controversy concerns the degree to which the function of 'visual' cortices is similar in people who are blind and sighted. This controversy speaks both to the developmental origins of categorical perceptual representations in vOTC as well as to broader questions about the role of experience and innate predispositions in brain development. The controversy arises most prominently in the case of vOTC, where there are strong claims of preservation in blindness on the one hand [18,19] and equally strong claims of drastic change in function on the other [1].

There is a range of perspectives on this point. According to one perspective we will call the 'similar representation view' - that only the sensory modality that drives vOTC changes in blindness, from vision to touch and audition. In blind and sighted people, the vOTC plays the same behavioral role (shaped-based object recognition), similar representations (of object shape) and similar domain-specific organization, irrespective of visual experience. This view is supported by studies showing selective responses to faces, places and written text in blind individuals in similar regions to the FFA, PPA, and VWFA in sighted people [17-20]. On this view, these regions are involved in shape-based recognition of faces, places and words even in people born blind (e.g., [26]).

An alternative possibility is that ventral visual areas have related but substantially different functions in blind and sighted people. On the 'related representation hypothesis' the ventral stream switches its behavioral role and representational content, even while maintaining elements of domain-specific organization observed in people who are sighted. For example, in the case of the VWFA, it may be involved in visual letter/word recognition in the sighted and high-level language processing in the blind [27]. In both cases, the representations are language-relevant, but the cognitive content (letter shapes vs. sentence structure) and behavioral role (word recognition vs. language comprehension) is quite different. Likewise, the FFA may respond to high-level social information in people who are blind and while performing face recognition in people who are sighted [19]. An intermediate alternative is that these regions maintain their behavioral role (e.g., person recognition) but do so with different representations (e.g., of voices).

**Competing hypotheses:** The key issue regards which features of the ventral 'visual' system is preserved in blindness.

1. If innate predispositions determine cortical function, then the ventral 'visual' stream areas in congenitally blind individuals perform the same function of object perception and represent the same type of object features as in sighted people (*similar representation view*).

2. If cortical function is modifiable by experience, then ventral stream object-perception 'visual' regions may serve a different behavioral role and will contain different representations (e.g., perceptual vs. conceptual) across sighted and blind people (*related representations hypothesis*).

**Proposed approach for resolution:** Previous work in this area has been conducted by different research groups in parallel. Studies have examined either a) responses across different object categories to map domain-specific regions (e.g., [18,28]) or b) higher-cognitive functions like responses to language and number (e.g., [21,24]). This has resulted in accumulating parallel evidence for both

alternative interpretations, highlighting the controversy. The lack of consensus in this field is also likely due to the low prevalence of congenital blindness and the logistical challenges of testing this population. This scarcity hinders even more than typical a convergence toward a cohesive perspective of how the vOTC develops in the absence of vision. The goal of this adversarial collaboration is to bring together different groups to approach a resolution by testing both perspectives in the rare but important population of congenitally blind individuals.

The proposed approach is to lay out alternative theoretical possibilities and mutually agreed upon predictions of the different theoretical views for the function of the vOTC across blind and sighted individuals. We will use fMRI, behavior, and computational modeling to characterize and test alternative hypotheses about the representational contents of vOTC across blind and sighted populations. Data analytical pipelines and their implementational details will be agreed to by all collaborators.

First, we will determine whether the same vOTC regions respond during haptic/audio perception also respond strongly to language and other higher-cognitive functions. Once the category-selective yet "cross-modal" regions are localized within vOTC (e.g., faces, places, words), we will investigate the representational content within each domain by creating stimuli that are maximally controversial across computational theories (in the style of [29]). These computational models will employ approaches that stipulate physics-based, common representations (e.g., [30]) as well as multisensory deep neural networks [31,32]. We will then employ representational similarity analysis (e.g., [33]) to contrast consistency between the predicted similarity structures and what we measure in the brain across blind and sighted populations. This will unconfound lower-level perceptual and higher-level conceptual features that could be represented in these regions.

For example, an FFA-like area that responds in a blind group during haptic face perception might code the configuration of shape-related facial features, like in the sighted. Alternatively, the FFA may be involved in a similar computation (i.e., person recognition) but via other features, such as voice. If this is the case, the representational structure should reflect acoustic rather than shape-based features. A third possibility is that for people who are blind, the FFA represents abstract social information, such as person identity or self-relevance, akin to representations of the ventromedial prefrontal cortex. If this is the case, then despite responding to faces in some tasks, the FFA in blind people would not be specifically involved in person perception. Likewise, does a PPA-like area in people who are blind represent the spatial layout of a scene, as in sighted people? Alternatively, does it represent the acoustic features relevant to auditory scene recognition? Finally, does it code semantic features, such as indoor vs. outdoor, or manmade vs. natural classifications?

**Concrete outcomes:** One goal of this adversarial collaboration is to jointly articulate theoretical proposals and their predictions across research groups. The next goal is to test these predictions. The proposed experiments will map out the function and informational content of the ventral 'visual' stream across sighted and blind individuals. This will include the first computational models of object perception and representations in blindness. We believe this collaboration will resolve a bottleneck in this area of research by creating dialogue and mutual collaboration across laboratories with different perspectives. By integrating computational methods and different theoretical perspectives, we hope to resolve the current controversy. We will further make the neuroimaging data publicly available, pending regulatory body and participant approval, which will facilitate further investigations and analysis of what will be a rich dataset. Description of analytical pipeline and analyses scripts will also be shared publicly upon publication submission. This will ensure that conclusions are robust to different analytical pipelines.

**Benefit to the community:** The findings would not only resolve this debate, but would also have broader implications more broadly for theories of neurodevelopment and plasticity. The project will aid in clarifying the role of visual experience in shaping the human brain and mind, which is relevant broadly across different fields including neuroscience, psychology and philosophy communities. This initiative is an opportunity for international leading researchers in the fields of object perception and neuroplasticity to collaborate in addressing a key issue in cognitive neuroscience. We also hope to inspire other researchers with differing theoretical leanings to collaborate rather than working in parallel, as we believe this is critical to advancement in any field.

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

feedback from CCN community, organize workshop, write position paper, attend CCN, conduct experiments.

*N Apurva Ratan Murty* (Massachusetts Institute of Technology) will develop new theory, organize workshop, write position paper, attend CCN, conduct experiments.

*Stefania Mattioni* (Université Catholique de Louvain) will develop new theory, organize workshop, write position paper, attend CCN, conduct experiments.

**Statements of commitment:** All members of the committee agree to the GAC initiative, including: 1) incorporating community feedback and potentially welcoming new members; 2) running the kickoff workshop during CCN2021; 3) writing the position paper; 4) attending and presenting progress at CCN2022.

