# OpenReview forum: "How does visual experience shape representations and transformations along the ventral stream?"
_ccneuro.org/CCN/2021/Workshop/GAC_

### Official Review · ~Adrien_Doerig1 · 2021-07-23
**A very good proposal aiming to address the impact of experience on visual representations**

**Rating:** 9
**Confidence:** 4

**Review:**

How, and to what extent experience shapes the brain is a central debate in neuroscience. Here, Bedny et al. propose a promising adversarial collaboration to tackle this question in the context of visual representations in congenitally blind subjects. As they describe in their literature review, this is a very timely topic that is still unresolved.

There are two opposing views. The similar representation view holds that innate predispositions determine cortical functions. Under this view, the representations in congenitally blind subjects should be determined innately and therefore not differ from the representations of sighted subjects. In contrast, the related representation view holds that cortical function is modified by experience, so that representations in blind subjects should differ from those of sighted subjects. To resolve this debate, the authors will experimentally assess which of these views is most promising by characterizing representations of blind subjects using imaging and computational methods.

I find this subject important, and the proposal is well in the spirit of adversarial collaborations, with two opposing sides wishing to work together to progress. I strongly recommend accepting this submission.

---

### Official Review · ~Xiaoxuan_Jia1 · 2021-07-26
**A specific proposal addressing whether the visual representation is hardwired or experience dependent**

**Rating:** 9
**Confidence:** 4

**Review:**

To what degree visual experience can shape representation is a key question in neuroscience. The proposal has identified a very specific question to address this general debate: "the degree to which the function of ‘visual’ cortices is similar in people who are blind and sighted". Including people from both camps of the debate and people with diverse background, they propose to work together in synchrony and use fMRI, behavioral and computational modeling tools to address this question. This is a well-defined and concrete proposal.